# Interventional Endoscopy for Palliation of Luminal Gastrointestinal Obstructions in Management of Cancer: Practical Guide for Oncologists

**DOI:** 10.3390/jcm11061712

**Published:** 2022-03-19

**Authors:** Matthew Kim, Mandip Rai, Christopher Teshima

**Affiliations:** Division of Gastroenterology and Hepatology, Department of Medicine, The Center for Advanced Therapeutic Endoscopy and Endoscopic Oncology, St. Michael’s Hospital, Toronto, ON M5B 1W8, Canada; mattbskim@gmail.com (M.K.); mandip.rai@queensu.ca (M.R.)

**Keywords:** gastrointestinal endoscopy, stents, colorectal cancer, esophageal cancer, gastric outlet obstruction

## Abstract

Self-expanding metal stents placed during endoscopy are increasingly the first-line treatment for luminal obstruction caused by esophageal, gastroduodenal, and colorectal malignancies in patients who are not candidates for definitive surgical resection. In this review, we provide a practical guide for clinicians to optimise patient and procedure selection for endoscopic stenting in malignant gastrointestinal obstructions. The role of endoscopic stenting in each of the major anatomical systems (esophageal, gastroduodenal, and colorectal) is presented with regard to pre-procedural patient evaluation, procedural techniques, clinical outcomes, and potential complications, as well as post-procedure aftercare.

## 1. Introduction

Endoscopic insertion of self-expanding metal stents (SEMSs) is at the forefront of management of malignant obstruction of the gastrointestinal tract (GIT). In both palliative and neoadjuvant settings, stenting plays a vital role in alleviating symptoms related to esophageal, gastroduodenal, or colorectal obstruction. Due to their safety profile and ability to provide rapid symptomatic relief, SEMSs have largely eclipsed radiotherapy and surgery as the first-line treatment in these scenarios. Pre-procedural assessment often involves comprehensive interdisciplinary discussion and work-up between relevant specialties, such as clinical and radiation oncologists, radiologists, surgeons, and gastroenterologists. Once the patient has been carefully selected as a candidate to benefit from the procedure, the SEMS placement is predominantly performed by trained interventional endoscopists, who utilise fluoroscopy and ultrasound to safely deploy the stents in the relevant anatomical position. In this review, we aim to provide a practical guide for physicians regarding the current role of endoscopic intervention in management of non-biliary malignant gastrointestinal obstruction. These procedures are discussed with respect to appropriate patient selection, technical considerations, clinical outcomes, and complications, as well as post-procedural management and future directions.

## 2. Stenting in Upper Gastrointestinal Malignancies

In parallel with the advances in interventional endoscopy in recent years, SEMSs have become an established forefront treatment strategy for alleviating obstructive symptoms in those with esophageal, gastroduodenal, or pancreatobiliary malignancies that have progressed to compromised luminal patency. The role of SEMSs has now further broadened to not only treat malignant strictures but also malignant fistulae, perforations, and external compressions. As such, the predominant indication for insertion of SEMS in the GIT is palliation of malignant obstruction. Many patients with upper GIT malignancies present with advanced inoperable disease related to the distensile nature of the GIT, which predisposes patients to be asymptomatic until the lumen is severely compromised [1]. This is further compounded by the fact that most GI cancers affect the elderly with various comorbidities, which renders them poor surgical candidates.

Esophageal cancer is one of the chief causes of cancer-related mortality and its five-year survival rate is less than 10% [2,3]. Over 70% of affected patients present with dysphagia, with many of them having unresectable disease at the time of diagnosis and eventually developing esophageal obstruction [4,5,6].

Gastric and pancreatic cancers are now amongst the leading causes of cancer-related mortality. Almost 40% of patients with gastric cancer have inoperable disease whereas up to 85% of patients with pancreatic cancer have unresectable tumours [7,8]. Many of these patients develop gastroduodenal outlet obstruction (GOO) over the latter stages of their disease processes, with pancreatic cancer being the most common cause of malignant GOO in Western populations and gastric adenocarcinoma being the most frequent cause in Asian populations [9]. GOO can also be secondary to infiltration or compression from metastatic lymph nodes, biliary/ampullary tumours, and other solid-organ malignancies. Nasogastric tubes and percutaneous gastrostomy are of limited value in these situations, as the location of the obstruction is anatomically distal to the site of intervention of these measures. Although a useful approach, surgical gastroenterostomy is also losing its popularity as a first-line intervention as it is associated with higher perioperative morbidity in comparison to SEMSs (whether it be traditional luminal stenting or EUS-guided gastrojejunostomy) [10].

## 3. Stenting in Colorectal Malignancies

Endoscopic stenting is indicated for patients with malignant colorectal cancer who are not surgical candidates and who have both clinical symptoms suggestive of obstruction and imaging findings consistent with obstruction. In select situations, endoscopic stenting for potentially curable left-sided obstructing colon cancer, as a bridge to surgery rather than emergent surgery, is an option. Some patients with extracolonic tumours causing large bowel obstruction can be considered for endoscopic stenting. However, success rates are lower and reintervention is more frequently required in this latter group compared to patients with colorectal cancer [11].

## 4. Pre-Procedural Patient Evaluation

The presentation of the patient differs in relation to the extent of the primary disease and tumour stage in conjunction with their baseline comorbidities. Whether the patient is an appropriate candidate for the procedure should be evaluated by a multidisciplinary team that considers the history and investigations alongside the patient and their family’s personal situation and expectations. All the relevant investigations and clinical information, such as radiologic imaging and latest endoscopic assessment, should be reviewed to devise delivery of the most effective palliative therapy in the least invasive approach possible [12].

Imaging such as fluoroscopic contrast studies and CT also plays a crucial role in defining the location and the morphological characteristics of the stricture and local/distant involvement. With the exception of when the passage of the scope is prohibited by high-grade complete obstructions, endoscopy can provide lesion visualisation and tissue sampling to delineate the degree of involvement of the malignancy. Biopsies of an obstructing tumour are recommended for pathological confirmation, but tissue confirmation is not mandatory.

The location of the stricture is highly relevant. In high esophageal tumours, care must be taken to ensure the proximal end of the SEMS is at least 2 cm below the upper esophageal sphincter to avoid airway compression and pain. For GOO, insertion of SEMS can impair further tissue acquisition (in particular with pancreatic cancer) and may cause biliary obstruction if the SEMS is deployed across the ampullary orifice. Similarly, whereas endoscopic stenting has been well-studied and proven to be effective in left-sided obstructions, distal rectal lesion stenting (within 5 cm of the anal verge) should be performed cautiously as it can lead to severe tenesmus and pain. In our experience, SEMS insertion for low rectal cancer is successful as long as the lesion is not too close to the anal–rectal verge. Proximal, right-sided colonic obstructions are technically more difficult to stent and should be assessed on a case-by-case basis.

The risk of stent migration can increase with tumour shrinkage post chemotherapy and radiotherapy and, depending on the clinical circumstances, stent insertion may be more appropriate after rather than before these treatments. Temporary removable stents have a role in these situations as they can allow stent upsizing as the luminal diameter increases post cancer treatment. Radiotherapy can also exacerbate the likelihood of stents causing perforation and erosive complications [13]. Another consideration is that the presence of a stent can impede accurate radiotherapy by widening the radiation field. The degree of stenosis is another factor to consider, as a partial obstruction that allows easy passage of the scope is associated with a high risk of stent migration, which is why we often choose to not place a stent in this circumstance. Contraindications to stent insertion include disease that is considered curative by surgery, terminally ill patients, uncorrectable coagulopathy, free gastrointestinal perforation, and sepsis [1]. Finally, the European Society of Gastrointestinal Endoscopy (ESGE) recommends against endoscopic stenting while patients are receiving antiangiogenic therapy such as bevacizumab, as there may be a higher risk of perforation in this setting [14,15,16].

## 5. Procedural Techniques

### 5.1. Bowel Preparation

Patients with esophageal obstruction and GOO are typically fasted for 12–24 h prior to the SEMS insertion. In those with GOO, insertion of a nasogastric tube in the days preceding the procedure is often necessary to ensure adequate decompression of the stomach prior to the endoscopy. For colonic obstruction, the use of oral bowel preparation is generally contraindicated given the risk of perforation. However, preparation with an enema to clean the colon distal to the stricture helps to facilitate visualisation and stent placement [17]. Tap water enemas can be administered to clear stool below the obstruction.

### 5.2. Stent Selection

A wide variety of SEMSs are now available to cater to different clinical circumstances. Uncovered and partially covered stents are less likely to migrate (0–6%) [5] but have higher rates of stent tumour ingrowth (17–36%) [13] and are considered permanent, as removal is difficult, if not impossible. In contrast, fully covered stents have the advantage of being removable and have virtually no risk of tumour ingrowth, but have a high stent migration rate (12–36%) [18,19]. However, these are not commonly used in GOO or colonic obstructions. We usually select a stent with a length adequate to traverse the entire length of the stricture and extend 1–2 cm further both proximally and distally. Long strictures can be overcome by insertion of more than one stent with stent overlap. The correct selection of stent length is also important to account for the curved gastroduodenal/colorectal anatomy to avoid the ends of the stent compressing directly against the wall of the lumen, as this may result in failure of the stent to achieve luminal patency or perforation.

### 5.3. Endoluminal Insertion of SEMS

Endoscopic stenting should only be performed by an operator who has competence in both endoscopy and fluoroscopy use. Experience is key as a study showed improved technical and clinical success when the operator has performed at least 30 endoscopic stenting procedures [20]. In addition to an appropriately trained endoscopist, experienced nursing staff, anaesthetists, and radiographers are necessary for patient sedation and monitoring, fluoroscopy, and resuscitation, if necessary.

SEMSs are typically inserted while the patient is under sedation with use of X-ray for fluoroscopic guidance. The scope is advanced to the level of the obstruction orally for esophageal obstruction and for GOO and via the anus for colonic obstructions. The ability to pass the scope through the stricture is not necessary for endoscopic stenting; if the scope can advance through the stricture, it often means the patient is not sufficiently obstructed and the risk of stent migration will be higher. Fluoroscopy is the cornerstone of successful stent deployment in these scenarios for safe guidewire advancement and identification of precise stent location. Under direct endoscopic and/or fluoroscopic visualisation, a guidewire is passed through the endoluminal obstruction using various catheters as necessary. Contrast is injected through the catheter with fluoroscopy used to characterise the stricture (Figure 1). The length of the stricture is important as, generally, the shorter the stricture (<5 cm), the easier it is to successfully place the stent [21].

Once the guidewire has been confirmed to traverse the length of the stricture, SEMSs are usually deployed over the guidewire again under direct endoscopic and fluoroscopic guidance. Final stent position and its patency can be evaluated with the use of contrast and fluoroscopy. Although malignant stricture dilation before or after stent placement is usually avoided due to the risk of perforation [22,23], we have found that for extremely high-grade strictures, gentle dilation using balloon dilators over the guidewire prior to introduction of the SEMS deployment device can further enhance successful stent deployment. It should be noted that when performing colonoscopy for colonic obstructions, as little CO_2_ insufflation as possible should be used as there is a potential risk of perforation proximal to the obstruction with over-distention of the bowel.

In cases where a previously deployed stent has become occluded secondary to tumour ingrowth or reactive tissue hyperplasia, the approach is usually similar. Guidewire access through the lumen of the existing stent followed by coaxial insertion of a new SEMS within the existing stent is performed to re-establish luminal patency (Figure 2). Fully covered stents sometimes have a role in this scenario as they minimise the risk of re-occlusion from tumor or tissue overgrowth.

### 5.4. Endoscopic Gastroenterostomy

Since being first described by Binmoeller in 2012 [24], endoscopic ultrasound-guided gastroenterostomy (EUS-GE) has fast gained acceptance as an alternative to traditional SEMS in relieving GOO. There are numerous variations of the technique, but all utilise an echoendoscope for sonographic visualisation of an unaffected segment of small bowel in close proximity to the stomach, followed by the placement of a lumen apposing metal stent (LAMS) from the stomach to this portion of small bowel. In our centre, we favour a variation in the traditional anterograde gastroenterostomy: we advance a guidewire beyond the obstruction to introduce devices such as a balloon or nasobiliary catheter through which we inject contrast and dye for confirmation of the endosonographically identified bowel loop as an appropriate loop distal to the level of obstruction. A transgastric puncture is then performed into this segment of the small bowel, followed by insertion of a short SEMS to create a fistula between the stomach and the small bowel, thereby bypassing the area of obstruction (Figure 3). The stent must be fully covered to prevent leakage of gastric contents into the peritoneal cavity. Although there are many suitable LAMS for this procedure, Hot-AXIOS (Boston Scientific, Marlborough, MA, USA) stents reduce the number of steps required by incorporating the transgastric puncture and stent deployment apparatus into a single device. Given the potential risk of causing peritonitis with this procedure, patients are given peri-procedural antibiotics.

## 6. Clinical Outcomes and Adverse Events

The procedure is considered technically successful if the SEMS has been effectively deployed across the stricture, whereas the procedure is considered clinically successful if there is subsequent resolution of obstructive symptoms that does not necessitate additional therapies.

### 6.1. Esophageal

Esophageal stent placement is technically successful in 97–100% of cases, with impressive and rapid symptomatic relief being reported in 83–100% of patients [13]. Covered stents have also been reported to have a high clinical success rate in the palliative management of esophageal fistulae [25,26]. Although it has been established that SEMS provide acute relief of malignant dysphagia with extremely high success rates, recurrent dysphagia within 4 to 10 weeks after stenting occurs in up to 50% of patients [4]. In comparison, radiotherapy exceeds at relieving long-term dysphagia but the onset of its beneficial effects is slow, and may even worsen the dysphagia due to radiation-induced swelling at the start of treatment. The complementary effects of immediate relief from stent insertion and long- term effect of brachytherapy make the combination of palliative stenting and radiotherapy an attractive option. Figure 4 demonstrates rapid improvement of the stricture immediately post stent insertion.

Immediate or early complications include chest pain, aspiration, haemorrhage, and perforation. Chest pain is one of the commonest complications, affecting 12–14% of patients, whereas hemorrhage occurs in 3–8% of patients [5] and usually does not require intervention [1]. Perforation is uncommon. For esophageal stents traversing the gastroesophageal junction, gastroesophageal reflux disease (GERD) is an extremely common issue and is reported in 70–100% of the patients post insertion of esophageal SEMS [12]. Unfortunately, stents designed with anti-reflux mechanisms have not consistently shown significant reductions in GERD in comparison to conventional SEMS in randomised controlled trials [27].

As mentioned previously, tumour in-growth is more likely to occur with uncovered stents, whereas stent migration is a more significant concern with covered stents [28]. Partial stent migration can be addressed with insertion of a second stent overlapping the primary stent. In case of complete migration, the old stent may be removed surgically or endoscopically if causing symptoms and a new stent inserted. Recurrent obstruction secondary to tumour ingrowth/overgrowth can usually be treated with insertion of a second stent within the existing stent. Problematic granulation tissue and epithelial hyperplasia can respond to balloon dilation, debulking therapy or re-stenting. Other late complications include stent torsion or fracture and stent erosion into the luminal wall causing fistulas.

Esophageal stents can cause airway compression (more frequent when stents are placed in the upper third of the esophagus). Proximal esophageal stenting is also associated with a higher risk of perforation due to the close proximity of anatomical structures.

### 6.2. Gastroduodenal

In one systematic review of SEMS insertion for GOO, the technical failure rate was found to be 3% whereas the clinical success rate was 87% [29]. Overall, gastroduodenal stenting is safer than esophageal stenting and is usually better tolerated by patients, with significantly less incidence of postprocedural pain. The most common complications are intestinal ulceration and bleeding caused by stent abrasion. Bleeding and perforation are more frequent in pancreatic than gastric malignancies: one systematic review of prospective studies found the perforation and bleeding rate to be 1.2% and 4.1%, respectively [30]. Similar to esophageal stenting, stent migration occurs less frequently with uncovered stents. However, recurrent obstruction due to tumour ingrowth occurs in up to 50% of patients [31]. Ampullary obstruction and jaundice can be caused by insertion of gastroduodenal stents, especially when covered stents are used [32]. Fluoroscopy-guided percutaneous decompression is recommended in these situations as transpapillary endoscopic approaches via ERCP are usually impossible (Figure 5). However, novel EUS-guided biliary drainage procedures may still be possible depending on the stent position and the patient’s anatomy. When the duodenal stent is expected to lie across the ampulla, prophylactic biliary stenting may avoid these issues.

Surgical gastrojejunostomy (SGJ) is associated with symptomatic relief in 80–96% of patients with GOO [9,33]. Endoluminal SEMS have comparable rates of clinical success but lower complication rates while achieving more rapid relief of symptoms and earlier resumption of oral intake, which facilitates earlier initiation or resumption of chemotherapy than those who undergo SGJ [34]. However, as expected, SGJ has considerably less need for reintervention as tissue ingrowth and tumour obstruction are minimal causes of concern afterwards [35]. As such, for patients with a life expectancy of 3 months or longer, SGJ, with its better long-term results, has often been considered the treatment of choice. In comparison, EUS-GE appears to have similar long-term results to surgical gastrojejunostomy, but lower complication rates and faster post-procedural recovery. In one of the first case series including 10 patients, both the technical and clinical success rates were reported to be 90% [36]. Subsequent larger studies reported technical success between 90% and 92% and clinical success of 85–90% [31,37]. A study comparing enteral stenting to EUS-GE found no differences between clinical success or adverse events, but EUS-GE was found to have longer stent patency [38]. Similarly, in another retrospective study, symptom recurrence and re-intervention were significantly lower in the EUS-GE group than in the enteral stent group (4.0% vs. 28.6%) [38]. In a study in which EUS-GE was compared to SGJ, no significant differences were noted with regard to technical or clinical success rates [39]. Of note, all the patients in the EUS-GE group were considered non-surgical candidates, which further highlights the utility of EUS-GE in these scenarios as a highly effective alternative to SGJ, having comparable clinical success and fewer associated adverse events [40]. Finally, in a systematic review comparing EUS-GE with enteral stenting and SGJ, EUS-GE had significantly improved outcomes in comparison to enteral stenting while being associated with a shorter length of hospital stay, with no difference in rate of clinical success or adverse events in comparison to SGJ [41]. As such, EUS-GE has become the preferred approach at our centre for relief of malignant GOO in patients expected to live more than 3 months, whereas we continue to place gastroduodenal SEMS when patients have shorter life expectancies.

### 6.3. Colorectal

In a systematic review of eighty-eight articles, the median rates of technical and clinical success were 96% and 92% respectively [42]. Definitions of clinical success varied among the studies, but all included colonic decompression with resolution of obstructive symptoms within 72 h of stent placement. An example of a successful stenting of a malignant sigmoid stricture is seen in Figure 6. The most common adverse event following endoscopic stenting is stent migration, and the most serious complication is perforation, which may be immediate or delayed. Bleeding and pain may also occur. In the same systematic review, perforation occurred in 4.5% of patients and stent migration occurred in 11%. Stent migration occurs more frequently when the stricture is not completely obstructive, when the stent size is not optimal (too small or too short), and after chemotherapy or radiation when the tumour may shrink. Other issues include stent occlusion due to tumour ingrowth, in which case another stent can usually be placed within the original stent to relieve the obstruction. Failure of endoscopic stenting to relieve obstruction can occur due to several reasons, such as an additional site of more proximal obstruction from a synchronous lesion or extrinsic compression, stent obstruction from impacted stool, early stent migration, or technical failure due to poor stent positioning or incomplete stenting of the entire length of the stricture.

## 7. Aftercare and Nutrition

In the majority of cases, we perform SEMS placement in an outpatient setting and the patients are discharged on the same day following the procedure. Hospital inpatients are typically observed overnight and then discharged the following day. Patients are advised with regard to management of transient chest pain related to SEMS expansion for esophageal stents. They also need to be informed of potential early and late complications and the signs and symptoms that herald these complications (Table 1). Patients with endoluminal SEMS insertion should be educated to adhere to a low-residue diet to reduce risk of food impaction. Typically, patients are advised to fast for 12 h post stent insertion to monitor for immediate complications (namely bleeding or perforation), followed by either a clear or full-fluid diet, followed by a progression to soft foods with avoidance of fibre. Patients with esophageal SEMS are recommended to eat in an upright position and are often prescribed proton-pump inhibitor therapy to reduce GERD. Patients with colonic stents are kept on regular stool softeners to prevent stent impaction with stool.

## 8. Future Prospects

The utility of endoscopically inserted SEMS is ever-evolving. Newer technology includes biodegradable stents that avoid long-term complications of permanent stents in benign disease and can serve as a temporary bridge to surgery. Drug-eluting or radioactive stents can reduce risk of recurrent obstruction in malignant disease [1]. A systematic review comparing the efficacy of stent insertion alone to stent insertion combined with any active oncological treatments in palliative cases of esophageal cancer concluded that irradiation stents may prolong survival and that stenting combined with oncological treatment does not increase the risk of complications compared to stenting alone [4]. Further studies are warranted to delineate what combination of treatments is effective.

## 9. Conclusions

Endoscopic insertion of SEMS has broad applications in the management of malignant luminal obstructions of the gastrointestinal tract. It has been established as a fast and effective method for relieving obstruction and restoring luminal continuity but it is not without complications, and careful patient selection is paramount to optimise clinical outcomes. Newer techniques such as EUS-GE have further expanded the role of endoscopic interventions that complement other oncological treatments in both palliative and non-palliative settings. Ongoing advances in therapeutic endoscopy are likely to parallel further development of novel strategies in management of malignant conditions for years ahead.

## Figures and Tables

**Figure 1 jcm-11-01712-f001:**
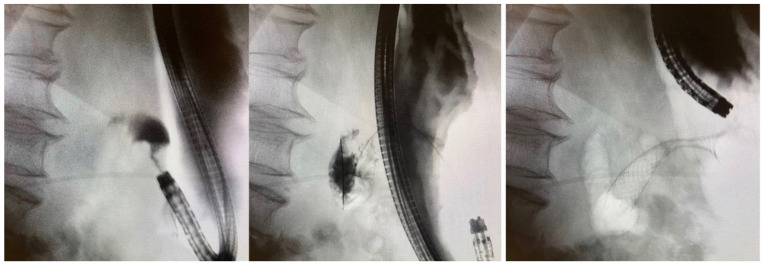
An 85-year-old woman with inoperable gastric cancer. Fluoroscopic images with contrast show presence of a short, high-grade stricture at the level of the pylorus/antrum (**left**). Guidewire advancement through the stricture into unaffected distal bowel segment to facilitate stent deployment (**center**). Successful deployment of uncovered SEMS across the area of stenosis (**right**).

**Figure 2 jcm-11-01712-f002:**
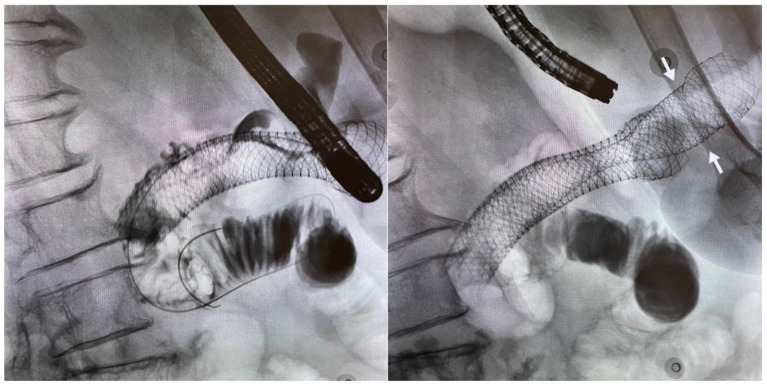
Same patient presenting with obstructive symptoms three months after first SEMS insertion. Note lack of contrast flow within the mid-portion of the existing stent due to tumour ingrowth (**left**). Deployment of second longer stent (arrows) within the existing stent with subsequent improvement in the patient’s symptoms (**right**).

**Figure 3 jcm-11-01712-f003:**
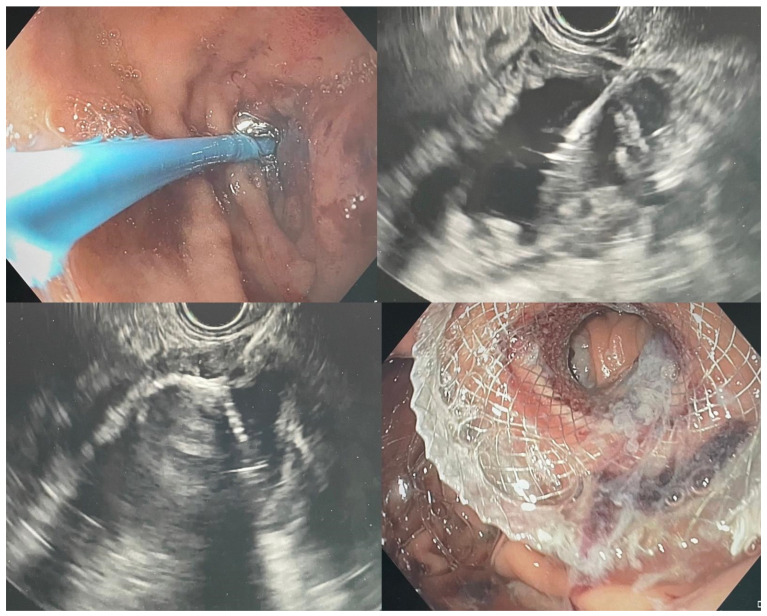
A 71-year-old woman with malignant GOO secondary to pancreatic cancer. Insertion of a catheter across the duodenal stricture to insufflate the distal small bowel with water and dye (**top left**). Puncture of the distended segment of small bowel under sonographic guidance for confirmation of adequate location (**top right**). Deployment of SEMS (Hot-Axios, Boston Scientific) under EUS guidance (**bottom left**). Creation of EUS-GE as seen on endoscopic view (**bottom right**).

**Figure 4 jcm-11-01712-f004:**
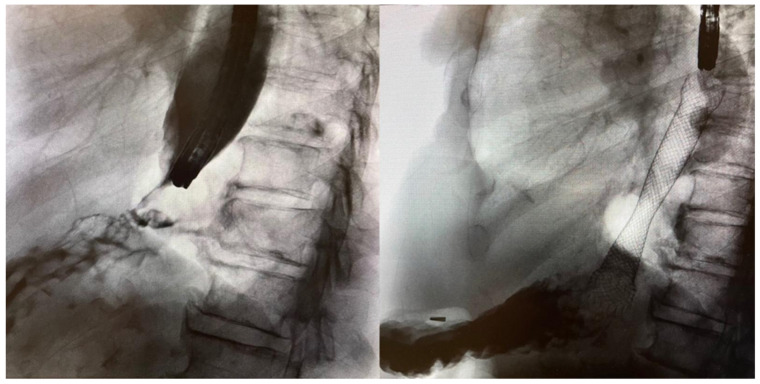
A 73-year-old man with esophageal cancer undergoing neoadjuvant chemotherapy presenting with worsening dysphagia. Presence of a short stricture at the gastroesophageal junction present on contrast aided fluoroscopic images (**left**). Successful deployment of a partially covered SEMS across the stricture with rapid flow of contrast seen to into the stomach post stent insertion (**right**).

**Figure 5 jcm-11-01712-f005:**
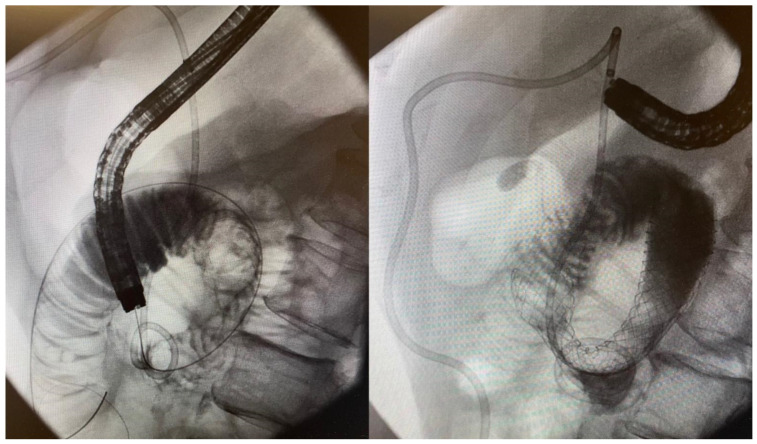
A 64-year-old male with cancer of the uncinate process of the pancreas who presents with both biliary and GOO. Long stricture in the second part of the duodenum seen on fluoroscopy (**left**). Insertion of uncovered enteral stent across the stricture. Note the presence of percutaneous biliary drain inserted prior to insertion of duodenal stent (**right**).

**Figure 6 jcm-11-01712-f006:**
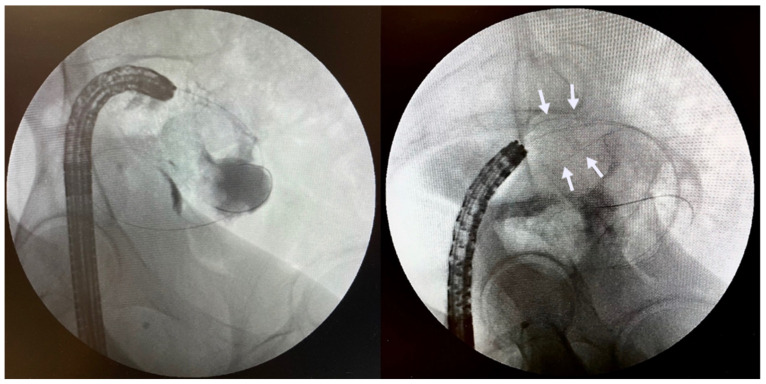
A 54-year-old woman with distal sigmoid obstruction due to obstructing tumour. Confirmation of colonic stricture with dilated proximal colon seen (**left**). Deployment of uncovered SEMS (arrows) for colonic stenting as a bridge to surgery (**right**).

**Table 1 jcm-11-01712-t001:** Periprocedural considerations for management of malignant strictures affecting particular anatomical location of the gastrointestinal tract.

	Esophageal	Gastric Outlet Obstruction	Colonic Obstruction
Type of stent	Uncovered in most circumstances Covered stents for fistulas and perforations	Uncovered Lumen-opposing stents in EUS-GE	Uncovered
Anatomical considerations	Ensure proximal end of stent <2 cm below UES for high esophageal strictures	Avoid stenting over ampulla if possible to avoid biliary obstruction	Avoid distal end of stent being in close proximity to anal verge
Pre-procedural management	Fasting 12–24 h prior	Fasting 12–24 h prior if not for days leading up to procedure May require NGT for decompression of stomach Antibiotics and discontinuation of anticoagulants/antiplatelet agents if for EUS-GE	Enema to clear colon distal to the level of obstruction; in case of partial obstruction may consider cautious use of oral bowel prep in select cases
Post-procedural recommendations	Anti-reflux medications for stents traversing the GEJ Low-residue diet	Low-residue diet	Low-residue diet Stool softeners
Post procedural complications to monitor	Retrosternal pain Gastroesophageal reflux Fistulation Perforation Stent migration Haemorrhage Tumor ingrowth	Biliary obstruction Abdominal pain Peritonitis Haemorrhage Perforation Tumor ingrowth Stent migration	Perforation Rectal pain for low-rectal strictures Haemorrhage Tumor ingrowth Stent migration

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
