# Peer review of "Interventional Endoscopy for Palliation of Luminal Gastrointestinal Obstructions in Management of Cancer: Practical Guide for Oncologists"

_jcm, 2022, doi:10.3390/jcm11061712_

Round 1

Reviewer 1 Report

Professor Kim submit a review on the endoscopic SEMS in management of malignant obstruction of the gastrointestinal tract. The manuscript shows many novel information about advances in therapeutic endoscopy and could open a new scenarios in the malignant GI luminal obstructions treatment. The figures summarizes well the results.
I believe this manuscript will give Journal of Clinical Medicine readers to understand development of novel strategies in management of malignant conditions for years ahead.

Author Response

Thank you.

  • the highlighted areas which correspond to previous publications have all been addressed and checked with online plagiarism checker 
  • abstract have now been included in the manuscript
  • keywords now also included included in the manuscript

Reviewer 2 Report

Very nice and practical review of the topic.

In two of the figures (2 and 6), the stent at deplyment is difficult to see and needs arrows on the figure - or delete the respective image. 

Author Response

Thank you.

  • Have included the arrows to better highlight the position of the stents within the figures 2 and 6. If the changes are still not adequate for figure 6 we will exclude the image for the manuscript

Reviewer 3 Report

Congratulations on this is a clear, well-written review of the current application of stents in GI palliation. It might be interesting background information: how many (or proportion of) patients get stents vs. RT or surgery in case of obstruction on each of mentioned levels at the moment, and has that trend changed over time.

Author Response

Thank you for the comment as it is indeed a point we should emphasize in the algorithm.

- Due to the logistical delays related to planning of radiotherapy as well as the clinical delay in response of symptoms post treatment, overwhelming majority of patients now undergo endoscopic stenting for rapid relief of their obstruction as the first-line (and often only line) of treatment. Similarly, surgery is usually only reserved for patients who have failed attempt at endoscopic stenting and is often declined by patients as a palliative measure. We have rephrased the manuscript to better highlight the above.